# Development and Validation of Indirect Enzyme-Linked Immunosorbent Assays for Detecting Antibodies to SARS-CoV-2 in Cattle, Swine, and Chicken

**DOI:** 10.3390/v14071358

**Published:** 2022-06-22

**Authors:** Abhinay Gontu, Erika A. Marlin, Santhamani Ramasamy, Sabarinath Neerukonda, Gayatri Anil, Jasmine Morgan, Meysoon Quraishi, Chen Chen, Veda Sheersh Boorla, Ruth H. Nissly, Padmaja Jakka, Shubhada K. Chothe, Abirami Ravichandran, Nishitha Kodali, Saranya Amirthalingam, Lindsey LaBella, Kathleen Kelly, Pazhanivel Natesan, Allen M. Minns, Randall M. Rossi, Jacob R. Werner, Ernest Hovingh, Scott E. Lindner, Deepanker Tewari, Vivek Kapur, Kurt J. Vandegrift, Costas D. Maranas, Meera Surendran Nair, Suresh V. Kuchipudi

**Affiliations:** 1Department of Veterinary and Biomedical Sciences, The Pennsylvania State University, University Park, PA 16802, USA; abhinay@psu.edu (A.G.); erika.marlin@pfizer.com (E.A.M.); sqr5895@psu.edu (S.R.); gxa5148@psu.edu (G.A.); jkm5870@psu.edu (J.M.); mjq5073@psu.edu (M.Q.); padmaja@psu.edu (P.J.); skc172@psu.edu (S.K.C.); nkk5370@psu.edu (N.K.); ska5899@psu.edu (S.A.); lcl122@psu.edu (L.L.); eph1@psu.edu (E.H.); 2Animal Diagnostic Laboratory, Department of Veterinary and Biomedical Sciences, The Pennsylvania State University, University Park, PA 16802, USA; rah38@psu.edu (R.H.N.); kmk6898@psu.edu (K.K.); 3Clinical & Diagnostic Assay Development Group, Pfizer, Pearl River, NY 10965, USA; 4U.S. Department of Health and Human Services, Silver Spring, MD 20993, USA; nnvsnath@gmail.com; 5Department of Chemical Engineering, The Pennsylvania State University, University Park, PA 16802, USA; czc325@psu.edu (C.C.); vqb5186@psu.edu (V.S.B.); cdm8@psu.edu (C.D.M.); 6Department of Integrative and Biomedical Physiology, The Pennsylvania State University, University Park, PA 16802, USA; aur1121@psu.edu; 7Huck Institute of Life Sciences, The Pennsylvania State University, University Park, PA 16802, USA; amm504@psu.edu (A.M.M.); rmr29@psu.edu (R.M.R.); sel27@psu.edu (S.E.L.); vxk1@psu.edu (V.K.); kjv1@psu.edu (K.J.V.); 8Madras Veterinary College, Tamil Nadu Veterinary and Animal Sciences University, Chennai 600007, India; drnpvel@gmail.com; 9Department of Biochemistry and Molecular Biology, The Pennsylvania State University, University Park, PA 16802, USA; 10Department of Animal Science, The Pennsylvania State University, University Park, PA 16802, USA; jrw140@psu.edu; 11Pennsylvania Department of Agriculture, Pennsylvania Veterinary Laboratory, Harrisburg, PA 17110, USA; dtewari@pa.gov; 12Center for Infectious Disease Dynamics, The Pennsylvania State University, University Park, PA 16802, USA; 13Department of Biology, The Pennsylvania State University, University Park, PA 16802, USA

**Keywords:** ELISA, serology, surveillance, SARS-CoV-2, cattle, swine, chicken

## Abstract

Multiple domestic and wild animal species are susceptible to SARS-CoV-2 infection. Cattle and swine are susceptible to experimental SARS-CoV-2 infection. The unchecked transmission of SARS-CoV-2 in animal hosts could lead to virus adaptation and the emergence of novel variants. In addition, the spillover and subsequent adaptation of SARS-CoV-2 in livestock could significantly impact food security as well as animal and public health. Therefore, it is essential to monitor livestock species for SARS-CoV-2 spillover. We developed and optimized species-specific indirect ELISAs (iELISAs) to detect anti-SARS-CoV-2 antibodies in cattle, swine, and chickens using the spike protein receptor-binding domain (RBD) antigen. Serum samples collected prior to the COVID-19 pandemic were used to determine the cut-off threshold. RBD hyperimmunized sera from cattle (*n* = 3), swine (*n* = 6), and chicken (*n* = 3) were used as the positive controls. The iELISAs were evaluated compared to a live virus neutralization test using cattle (*n* = 150), swine (*n* = 150), and chicken (*n* = 150) serum samples collected during the COVID-19 pandemic. The iELISAs for cattle, swine, and chicken were found to have 100% sensitivity and specificity. These tools facilitate the surveillance that is necessary to quickly identify spillovers into the three most important agricultural species worldwide.

## 1. Introduction

Severe acute respiratory syndrome coronavirus (SARS-CoV-2) has infected over 500 million humans and caused over 6 million fatalities worldwide as of 19 April 2022 [1]. While the precise origin of SARS-CoV-2, and its mode of introduction into the human population, is not yet fully resolved, there is evidence for that SARS-CoV-2 originated from bats [2,3,4,5]. As zoonotic viruses infecting humans can spill back into susceptible animal hosts [6], SARS-CoV-2 has demonstrated the ability to infect many non-human animal host species. The list of animal species susceptible to SARS-CoV-2 infection continues to grow and includes domestic animals, primates, pet animals, and zoo animals [7,8,9,10,11,12,13,14,15,16]. In addition, based on the ability of the spike protein to bind to the ACE-2 receptor, computational predictions have identified dozens of additional possible animal hosts for SARS-CoV-2 [17,18]. SARS-CoV-2 continues to evolve, and several novel variants have emerged since its first identification [19]. Experimental infection with the SARS-CoV-2 showed that livestock species, such as cattle and swine, are susceptible to the virus [20,21]. Respiratory ex vivo organ cultures of cattle and sheep sustain SARS-CoV-2 replication [22]. Experimental studies show SARS-CoV-2 spillover into animal hosts can lead to rapid adaptation and accelerated variant emergence, highlighting the extraordinary plasticity and adaptive potential of SARS-CoV-2 [23,24]. There are few, or no, experimental infection studies performed in cattle, swine, and chicken with emergent VOCs such as Delta and Omicron. With a recent report suggesting the possibility of the spillover of SARS-CoV-2 in cattle [25], a serological investigation for SARS-CoV-2 antibodies in livestock species is further necessitated. 

SARS-CoV-2 spillover and adaptation to livestock species can cause a significant impact on food safety and security, as well as both animal and public health. Therefore, targeted epidemiological investigations are necessary to monitor the potential SARS-CoV-2 infection of livestock species [26]. SARS-CoV-2 diagnosis in humans is primarily carried out using virus detection either by nucleic acid amplification tests (NAATs) or rapid antigen assays [27,28,29]. Experimental SARS-CoV-2 infection studies in livestock and other animal species show asymptomatic infection, and viral RNA was detectable for a limited period after infection [20,21,30,31]. This result demonstrates that there will be a short window for virus detection in animals for SARS-CoV-2 antigen or nucleic acid. SARS-CoV-2 infection of deer and cattle results in a detectable antibody response [9,11,20,31,32], and there is strong evidence that SARS-CoV-2-specific antibodies remain detectable for at least one year in humans [33,34]. Hence, serological screening of livestock for SARS-CoV-2 spillover is a rational and practical approach.

The recombinant SARS-CoV-2 spike receptor-binding domain (RBD) possesses multiple neutralizing epitopes and shares less sequence similarity with other human and animal coronaviruses [35]. RBD is used as the antigen in diagnostic assays, including indirect enzyme-linked immunosorbent assays (iELISAs), to detect serum antibodies against SARS-CoV-2 in humans [36,37,38]. We report the development and standardization of species-specific iELISAs for the serodiagnosis of SARS-CoV-2 infection in cattle, swine, and chicken.

## 2. Materials and Methods

### 2.1. Preparation of Recombinant RBD

Recombinant SARS-CoV-2/RBD antigen was produced as detailed earlier by Stadlbauer et al., 2020 [36]. The plasmid pSL1510, containing SARS-CoV-2 Wuhan spike RBD (pCAGGS-RBD), was kindly provided by Florian Krammer, Mount Sinai, USA. The plasmids were purified using a Qiagen HiSpeed Maxiprep Kit and transfected employing the Expi293 Expression System (Expi293F cells, Expi293 Media and the ExpiFectamine 293 Transfection Kit, Catalog # 14524, ThermoFisher, MA, USA) as per the manufacturer’s instructions. Expi293F cells were cultured in shaker flasks at 37 °C with 8% CO_2_ and 120–130 rpm. Before transfection, the cells were resuspended to 3 × 10^6^/mL of Expi293 Media. For transfection, the plasmid (1 µg/mL of culture volume) was diluted in Opti-MEM, and ExpiFectamine 293 reagent was diluted in Opti-MEM in a separate tube and incubated at room temperature for five minutes. Then the diluted ExpiFectamine 293 reagent was added to the diluted plasmid and incubated at room temperature for 15 min. The reagent/plasmid mix was slowly transferred to the cell culture and then incubated at 37 °C, 8% CO_2_ with 120–130 rpm. ExpiFectamine 293 Transfection Enhancer 1 and 2 were added to the transfection culture ~20 h post transfection. The supernatant was harvested by centrifugation at 4000× *g* for 20 min on the third day. Cell viability and concentration were monitored throughout the transfection to ensure that the culture remained in log phase growth. Culture supernatant was incubated with pre-equilibrated Ni-NTA (ThermoSci HisPur, catalog # PI88223, ThermoFisher, MA, USA) resin in PBS (0.5 mL of equilibrated Ni-NTA for every 50 mL of supernatant) at 4 °C for one hour on a nutator. The resin was applied to a gravity column and washed four times with 10 column volumes of wash buffer (57 mM NaH2PO4, 30 mM NaCl, 20 mM Imidazole). Protein was eluted from the resin with 4 column volumes of elution buffer (57 mM NaH2PO4, 30 mM NaCl, 235 mM Imidazole). Eluted protein was dialyzed in phosphate buffered saline (PBS) and snap frozen for storage at −80 °C.

### 2.2. Serum Samples

The serum samples (cattle and chicken) submitted to the Pennsylvania State University animal diagnostic laboratory (PSU-ADL) for routine diagnosis were used in the study. Swine serum samples were procured from South Dakota State University, Brookings, South Dakota. Serum samples submitted before December 2019 were used as negative controls (pre-pandemic samples). COVID-19 pandemic samples were collected from 2019 to 2021.

### 2.3. Raising Hyperimmune Sera

All animal care and sample collections were approved and performed in accordance with the guidelines of the Institutional Animal Care and Use Committee at Pennsylvania State University and Cocalico Biologicals, Inc. (see Ethics Statement). Three- to six-month-old cattle and swine and six-month old layer hens were used for hyperimmune serum production. Animals (cattle *n* = 3, swine *n* = 6, chicken *n* = 3) were administered four doses of the antigen (1 mg/dose in cattle, 0.1 mg/dose in swine and 50 µg/dose in chicken) emulsified with MontanideTM Gel 02 PR (Seppic, France) for cattle and swine and Freund’s complete/incomplete adjuvant (CFA/IFA) for chicken (CFA for primary vaccination and IFA for booster), intramuscularly at two-week intervals. Serum from cattle and swine were collected at two-week intervals to check for seroconversion. The animals were terminally bled 14 days after the fourth dose of RBD, and the hyperimmune serum (HIS) was stored at −80 °C. Eggs were collected from the hyperimmunized chickens, and immunoglobulin (Ig)Y was purified by affinity chromatography and stored [39]. The antibody titers in the hyperimmune serum or IgY were confirmed by a virus neutralization assay.

### 2.4. Virus Neutralization Test

The SARS-CoV-2 live virus neutralization (VN) test was performed as previously described [37,38]. Vero E6 cells (CRL-1586, ATCC, VA, USA) were grown in 96-well microtiter plates prior to the day of the test. Serum samples to be analyzed were diluted two-fold and tested in triplicate. Then, 50 µL of each serum sample was incubated with 100 tissue culture infective dose_50_ (TCID_50_) units of the SARS-CoV-2, USA-WA1/2020 (NR-52281-BEI Resources, VA, USA) virus at 5% CO_2_ at 37 °C for one hr. The virus–serum mixture was added to the cell monolayers and incubated for three days. Appropriate cell and infection controls were maintained. The cells were observed for cytopathic effects in triplicate wells, and the observations were recorded. The reciprocal of the highest dilution of serum showing at least 66.7% protection (two out of three wells) was defined as the VN titer of the serum.

### 2.5. Indirect ELISAs

Optimization of iELISAs, including the antigen concentration and serum dilutions, was performed as described by Salazar et al and Gontu et al [37,38] with minor modifications. A set of 96-well ELISA plates (Cat # 44240421, Thermofisher, MA, USA) were coated with 50 µL of RBD antigen (2 µg/mL in PBS) per well, sealed and incubated overnight at 4 °C. Plates were washed with PBS containing 0.05% Polysorbate 20 using hydroFLEX microplate washer (TECAN, Switzerland) to remove the unbound antigen. Then, 200 µL/well blocking buffer was added and incubated at 37 °C for one hour. Stabilguard immunoassay buffer (SG01-1000, Surmodics, MN, USA) was used as the blocking and dilution buffer for testing serum from cattle and chicken. The ELISA for testing swine serum used 5% non-fat dry milk powder in PBS containing 0.05% Polysorbate 20 as the blocking buffer and 2.5% non-fat dry milk powder in PBS containing 0.05% Polysorbate 20 as the dilution buffer. After blocking, the plates were washed thrice and incubated with serum samples (1:50, in their respective dilution buffers) for one hour at 37 °C. Later, the plates were washed and 100 µL of the respective secondary antibodies were added: anti-bovine IgG−Peroxidase (Cat # A5295, Sigma-Aldrich, MO, USA); anti-Pig IgG−peroxidase (Cat # A5670, Sigma-Aldrich, MO, USA); or anti-chicken IgY (IgG)−peroxidase (Cat # A9046, Sigma-Aldrich, MO, USA) at 1:10,000 dilution in their respective dilution buffers and incubated at 37˚C for one hour. Plates were washed and 100 µL of substrate (10 mL phosphate-citrate buffer (Cat # PP4809, Sigma-Aldrich, MO, USA) containing hydrogen peroxide and 3,3′,5,5′-Tetramethylbenzidine dihydrochloride (Cat # T3405, Sigma-Aldrich, MO, USA)) was added to each well. Following 10 min of incubation, the reaction was terminated with 50 µL 3N HCL and absorbance was measured at 450 nm using a BioTek, ELx800 microplate reader. The reciprocal of the highest dilution of serum, which resulted in an absorbance value above the pre-determined cut-off was taken as the titer of the sample.

### 2.6. Cut-Off Determination

The assay’s diagnostic optical density cut-off (absorbance at 450 nm) was determined by screening pre-pandemic serum samples (*n* = 40 for cattle, *n* = 40 for swine, *n* = 40 for chicken). The samples with absorbance above the cut-off were considered to be positive for anti-SARS-CoV-2 antibodies, and those with absorbance below the cut-off were considered negative. Hyperimmune serum was used as the positive control in the assays.

### 2.7. Statistical Analysis

The sensitivity, specificity, and accuracy of the iELISAs were calculated using the diagnostic test evaluation calculator available at: https://www.medcalc.org/calc/diagnostic_test.php (accessed on 20 March 2022). Correlation between the iELISAs and the VN was performed by Pearson correlation, and linear regression analyses were performed using GraphPad PRISM 9 for macOS v.9.3.1(350), San Diego, CA, USA.

## 3. Results

### 3.1. Determination of P/N Threshold

The P/N cut-off was determined based on the absorbance values of pre-pandemic serum samples. The mean OD450 values for cattle (*n* = 40), swine (*n* = 40) and chicken *(n* = 40) serum samples were 0.111, 0.167, and 0.134 and had standard deviations (SDs) of 0.043, 0.084, and 0.047, respectively. The cut-offs were determined based on the distribution of absorbance values of the pre-pandemic samples for each species. Cattle and chicken serum samples had SDs of 0.043 and 0.047, for which the cut-offs were computed as five times the SDs of the negative samples over the mean of the pre-pandemic samples. Swine samples had an SD of 0.084, for which the cut-off was determined as three times the SD higher than the mean of the pre-pandemic samples. The cut-off values for the iELISAs were 0.325, 0.419, and 0.368 for the cattle, swine, and chicken sera, respectively.

### 3.2. Validation of iELISAs

The known positive sera (HIS) from cattle (*n* = 3), swine *(n* = 6) and chicken (*n* = 3) were utilized to validate the iELISAs (Figure 1). The performance of each of the species-specific iELISAs was evaluated with serum samples (*n* = 150 each from cattle, swine, and chicken) collected during the COVID-19 pandemic (Figure 1). The sensitivity and specificity of the ELISA compared to the VNT were calculated using a two-way contingency table (Table 1). The iELISAs developed for cattle, swine, and chicken had 100% sensitivity and specificity, (Table 1 and Appendix A).

### 3.3. Correlation between VNT and iELISAs

We tested if the results from the iELISAs correlated with the titers from VN assays. Pearson’s correlation coefficients showed a strong correlation with the titers from iELISAs and their VN titers (cattle: r = 0.93; 95% CI: 0.75–0.98; swine: r = 0.89; 95% CI: 0.28–0.99; chicken: r = 0.99; 95% CI: 0.98–0.99). Additionally, linear regression analyses determined the goodness-of-fit for each of the iELISA results compared with their respective VN titers. The R^2^ values of the iELISAs were >0.79, re-confirming the strong positive correlation between the assays (Figure 2).

### 3.4. Cross-Neutralization of SARS-CoV-2 Variants by HIS Raised to Wuhan RBD

We used the RBD of the Wuhan strain of SARS-CoV-2 as the antigen for the iELISA assays. Reports show a reduced cross-neutralization of VOCs by the serum antibodies against SARS-CoV-2 (Wuhan, China) [40]. The neutralizing ability of the HIS generated in this study was tested against various SARS-CoV-2 VOCs. The results show a broad cross-reactivity of the hyperimmune serum against both homologous viruses (Wuhan) and heterologous viruses (SARS-CoV-2 VOCs) (Appendix A). Cattle showed reduced neutralizing indices against Beta (42.83%), Gamma (51.94%) and Omicron (8.53%) VOCs, whereas swine showed reduced neutralization against the Omicron VOC (46.95%) compared to Wuhan (Appendix A). Interestingly, hyperimmune sera raised in chickens had similar neutralization indices against all the tested VOCs (Appendix A).

## 4. Discussion

There have been multiple natural transmissions of SARS-CoV-2 from human to animals, animal to animals, and animal to humans [13,30,41,42]. Some evidence shows that novel SARS-CoV-2 mutants can emerge from infected animal hosts [23]. One of the leading hypotheses for the possible source of the recent SARS-CoV-2 Omicron VOC is an unmonitored animal population [43,44]. As SARS-CoV-2 remains widespread in the global human population, it presents a high risk of exposure and possible spillover into agricultural animals. A recent report stating the possibility of a spillover of SARS-CoV-2 into cattle strengthens this hypothesis [25]. The possibility of spillover into livestock is likely even higher in underdeveloped and developing countries. While the initial experimental SARS-CoV-2 (Wuhan) infection studies indicated a low susceptibility of livestock species [10,20,21,30,45], the susceptibility of livestock to recently emerged VOCs is largely unknown.

Cost-effective high-throughput specific assays are essential for the surveillance of livestock species for possible spillover of SARS-CoV-2. Serum antibodies persist for months following infection; therefore, detecting SARS-CoV-2-specific antibodies is a rational approach for identifying evidence of these spillover events. Retrospective serological investigation demonstrating SARS-CoV-2 antibodies in white-tailed deer promoted subsequent studies that found widespread infection of SARS-CoV-2 in white-tailed deer [7,11,14]. The identification of multiple independent SARS-CoV-2 spillover events into wild ungulates highlights the utility of serological surveillance of animals. Because of significantly greater human interactions, farmed livestock species have a much greater risk of exposure to SARS-CoV-2 than wild animals. There have been several examples of animal coronaviruses originating from human coronaviruses or sharing common ancestors as those of human-adapted viruses. For example, bovine coronavirus shares a common ancestor with human coronavirus OC43 [46], and swine acute diarrhea syndrome coronavirus (SADS-CoV) originated from HKU-2-related bat coronavirus [47,48]. Although a few ELISA assays for detecting SARS-CoV-2 antibodies in cattle, swine, and chickens have been described [20,49,50], they are either not species-specific and/or were not validated with known positive and known negative serum samples, which is critical for accurately determining test sensitivity and specificity.

The RBD antigen-based iELISAs developed in this study have 100% sensitivity and specificity. Out of all the test sera from cattle, swine, and chicken (*n* = 150 from each species) collected during the COVID-19 pandemic, none were positive for SARS-CoV-2 antibodies by either iELISAs or VNT. As the screening of livestock always involves the herd or population rather than individual animals, ELISA could be one of the most cost-effective and high-throughput diagnostic tests available. Moreover, the highly sensitive VNT is cumbersome when the sample size is large and involves biosafety level 3 (BSL3) containment [51]. None of the negative sera from cattle, swine, and chicken were detected as positive in iELISA and VNT, thus exhibiting 100% specificity and indicating ELISA could be a standalone assay as the confirmatory diagnostic. The presence of neutralizing epitopes, and the poor conservation of SARS-CoV-2 RBD among coronaviruses, makes the RBD the most suitable target for the ELISA [52]. Correlation analyses on HIS showed that our iELISAs could serve as good surrogates to the VN assays, which is consistent with our previous finding of a strong correlation between RBD-iELISA and VN titers in human plasma [37,38]. Another assay, surrogate VN ELISA, is also being used for such surveillance purposes [11,53,54]. However, this assay will not be cost-effective for surveillance activities involving a large sample number.

It is essential to note that we used a relatively small number of opportunistic serum samples collected from December 2019 to October 2021. Stratified random sampling of livestock, in places with a risk of spillover, should be carried out to determine the prevalence of natural SARS-CoV-2 infections of agricultural species. We were not able to test if the sensitivity and specificity of the ELISAs were affected when testing antibodies against various VOCs. However, HIS produced by immunizing RBD from Wuhan in the three species in our study could neutralize all major SARS-CoV-2 VoCs. This observation indirectly implies that the RBD antigen used in the ELISA necessitates the identification of antibodies from the emerging VOCs of SARS-CoV-2. Additionally, if needs be, the ELISA can be updated using the RBD from an emerging VOC in the event of occurrence of a higher number of mutations in the virus. In conclusion, SARS-CoV-2 RBD-iELISAs for cattle, swine, and chickens are valuable for monitoring livestock exposure to known and currently circulating SARS-CoV-2 VoCs.

## Figures and Tables

**Figure 1 viruses-14-01358-f001:**
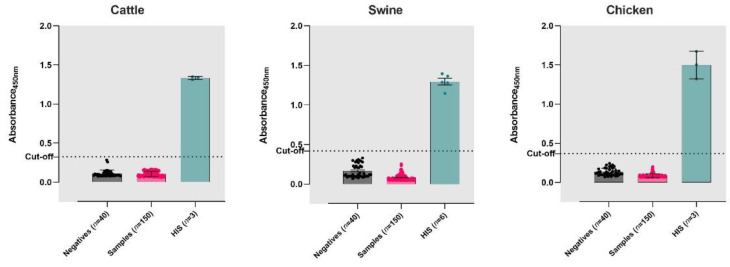
Performance of iELISAs on the serum samples collected from cattle, swine, and chicken. Pre-pandemic (known negatives) and pandemic-era serum samples from domestic animals were tested on species-specific iELISAs for cattle, swine, and chicken. The cut-offs of the iELISAs were determined by testing pre-pandemic sera from each species. Hyperimmune serum (HIS) samples were used as positive controls for the ELISAs. Bar heights represent the mean absorbance of the samples read at 450 nm. Error bars represent the standard deviation. Dashed lines indicate the cut-off for each of the assays.

**Figure 2 viruses-14-01358-f002:**
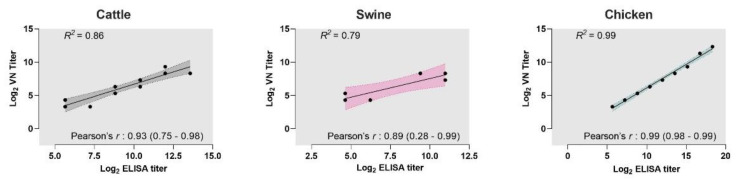
Correlation of cattle, swine, and chicken iELISAs with VNT. Correlation analyses were performed on the antibody titers from iELISAs and virus neutralization (VN) assays for each of the species: cattle, swine, and chicken. Strong correlation was observed between the titers from the assays (Pearson’s r: >0.8). Dots represent each sample of hyperimmune serum or IgY tested. Solid lines represent the linear regression curves of the titers of the samples analyzed by the assays. Shaded area indicates 95% confidence interval of the linear regression curve. R^2^ is the correlation coefficient from the linear regression curve.

**Table 1 viruses-14-01358-t001:** Estimation of sensitivity and specificity of iELISAs.

VNT Status	Total	Positive	Negative	Sensitivity/Specificity
**a. Cattle**				
Positive sera *	13	13	0	100% (13/13; sensitivity)
Negative sera	150	0	150	100% (150/150; specificity)
**b. Swine**				
Positive sera	6	6	0	100% (6/6; sensitivity)
Negative sera	150	0	150	100% (150/150; specificity)
**c. Chicken**				
Positive sera *	24	24	0	100% (24/24; sensitivity)
Negative sera	150	0	150	100% (150/150; specificity)

* Analyses conducted on serum samples from bleeds collected at multiple time points from three cattle, six pigs, and three chickens.

## Data Availability

The data from testing individual animal sera will be made available on request to the corresponding authors.

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
