# Peer review of "Development and Validation of Indirect Enzyme-Linked Immunosorbent Assays for Detecting Antibodies to SARS-CoV-2 in Cattle, Swine, and Chicken"

_viruses, 2022, doi:10.3390/v14071358_

Round 1
Reviewer 1 Report
This is a well written report on the development and validation of indirect enzyme-linked immunosorbent assays for diagnosis to SARS-CoV-2 in livestock.
The manuscript has a very solid content. The results of this study are likely to help control the SARS-CoV-2 from the point of view of One Health.
Author Response
Thank you for the supporting comments. Much appreciate the time you took to review the manuscript.
Reviewer 2 Report
This manuscript presents an operationally solid description of assays for detecting antibodies against the receptor binding domain of the SARS-CoV-2 spike protein for three food production species. The use of pre-pandemic samples from established sample banks provides an important benchmark for determining assay background signal. The study is sound overall as far as it goes. However, to support the claims made regarding assay sensitivity and specificity, additional elements that are not currently considered in the manuscript should be addressed:
- The positive controls (hyperimmune sera from immunized animals of each of the three species) appear to generate maximum OD signals likely to be the highest potential reading from a spectrophotometer (Figure 1). Are these signals within the linear range for the assay? What are the upper and lower limits of detection? Full assessment of these limits is important for understanding assay sensitivity and specificity, as it is currently not clear how the assays would perform with “borderline” samples such as those from animals several months following infection. Are these limits reproducible in separate assay runs done on different days?
- Do the sample banks contain sera from animals known to have been infected with other corona viruses (as described in lines 272-278)? Do these samples cross react with the SARS-CoV-2 RBD capture antigen in these assays?
- Given the reduction of neutralizing activity against VOCs for the hyperimmune sera (lines 239-247), what might be the impact of the changes in VOCs on ELISA sensitivity to detect prior exposure to these strains? This question should at least be considered in the Discussion.
Other comments:
- What were the criteria by which 5x SD was selected as the P/N cut-off for two of the assays, but 3x SD selected as the cut-off for the third assay?
- All abbreviations should be defined at first use.
- While the manuscript is clearly written, it should be reviewed carefully for agreement of subject and verb (plural vs singular). g., in line 86 “…domain…possess…” should be “…domain…possesses…”; in line 138 “…titers…was…” should be “…titers…were…”, among others.
Author Response
Comments and suggestions made by the reviewer to improve the scientific quality of the manuscript are appreciated by authors. All comments have been addressed, and authors’ responses to reviewer specific comments are provided in the attached document. The manuscript has also been revised incorporating the changes. Thank you for the consideration.
Please see the attachment.

Reviewer 3 Report
The paper presents some data obtained for the development and the validation of three species specific-indirect ELISAs (iELISAs) for livestock for detecting SARS-CoV-2 neutralizing antibodies. The authors decided to use the Spike protein receptor-binding domain (RBD) as antigen. The three target species are cattle, swine and chicken.
Such ELISAs are of outstanding importance for detecting easily the level of SARS-CoV-2 neutralising antibodies allowing to identify infection events. It has been demonstrated experimentally that some livestock species are susceptible to the virus. A recent report (Kerstin, 2022) suggest the possibility of SARS-CoV-2 spillover in cattle. As livestock is close to human, a spillover and adaptation of the virus to the livestock could have a major impact on food safety and security but also on animal and human health.
Materiel and methods
Lines 152-174: The authors shall explain if the results of these iELISAs are qualitative or quantitative and how they calculate the titer.
Results
Lines 191-193: On which scientific basis was determined the factor 5 or 3 to multiply the SD of negative samples to determine the cut-off value. The authors shall explain why this factor is different according to the species.
Lines 196-202: To validate their iELISAs, the authors used known positive sera, 3 sera for cattle and chicken and 6 sera for swine. However, in the Table 1, the sensitivity was determined by using 13 positive samples for cattle, 6 positive samples for swine and 24 positive samples for chicken. It is not clear at all for the reader where these numbers are coming from. As the 150 samples collected during the COVID-19 pandemic are all negative, the authors shall add some clarification about the origin of these positive samples.
The evaluation of the sensitivity has been done on a very few number of positive sera for these 3 ELISAs (i.e. 13, 6 and 24 respectively for cattle, swine and chicken). The authors shall test much more positive samples to validate such method in reliable and proper way. This could result in lack of sensitivity when using these iELISAs on a large scale.
Line 210: The authors shall clarify what they meant when speaking about “titers from iELISAs”. Do they correspond to the values of absorbance at 450nm or to some titers obtained after calculation?
Discussion
The authors shall improve the discussion by raising the issue and discussing about the small number of positive samples used to validate their iELISAs. This shall be done apart from the tests of antibodies against various VOCs.
Author Response

(The authors gave the same response as above.)

Reviewer 4 Report
The manuscript is well written and generally well organised. The authors present a novel iELISA against SARS-CoV-2 RDB spike protein to test the specificity and sensitivity in animal production.
There are a number of aspect that I would like to highlight:
Serum samples
The authors did not mention the sampling procedure. It is also avoided the ethict guideline in accordance with the University/institution involved in the study. Please, give further information about this point
Raising hyperimmune sera
- The autohrs did not mention the Ethics Guidelines about the hyperimmune sera production.
Please, provide information.
Line 137 "immunoglobulin (Ig)Y was purified by affinity chromatography", but the is neither a reference nor a description of the procedure. Please, add further information.
Minor revisions
Line 157 remove "the" before plates.
Line 158 add a comma (,) after "Then,"
Author Response

(The authors gave the same response as above.)

Round 2
Reviewer 2 Report
The authors have addressed the prior comments, either by editing or acknowledging areas where future work is planned to develop and validate the assays further.
Author Response
Thank you for the comments and suggestions. Much appreciate the time you took to review the manuscript.
Reviewer 3 Report
I would like to thank the authors to reply to all my comments/remarks and questions. The modified paper has been improved compared to the second version.
However, the explanations given by the authors do not reply at all to one of my comment. The authors shall reply to this comment prior publication.
Lines 191-193: On which scientific basis was determined the factor 5 or 3 to multiply the SD of negative samples to determine the cut-off value. The authors shall explain why this factor is different according to the species.
Author Response
Thank you for the comments and suggestions. Much appreciate the time you took to review the manuscript.
Authors' response: The cut-offs were determined based on the distribution of absorbance values of the pre-pandemic samples for each species as mentioned in the manuscript. Cattle and chicken serum samples had an SD of 0.04, for which the cut-offs were computed as five times SD of the negative samples over the mean of pre-pandemic samples. Swine samples resulted in an SD of 0.08, for which the cut-off was determined as 3SD over mean of pre-pandemic samples. Lines “195-200” in the revised manuscript.
Reviewer 4 Report
The authors have now incorporated the changes suggested as well as the information requested.
Author Response

(The authors gave the same response as above.)

Round 3
Reviewer 3 Report
I would like to thank the authors to reply to my last comment.